# GPI-80 Augments NF-κB Activation in Tumor Cells

**DOI:** 10.3390/ijms222112027

**Published:** 2021-11-06

**Authors:** Yuji Takeda, Yuta Kurota, Tomoyuki Kato, Hiromi Ito, Akemi Araki, Hidetoshi Nara, Shinichi Saitoh, Nobuyuki Tanaka, Norihiko Tsuchiya, Hironobu Asao

**Affiliations:** 1Department of Immunology, Faculty of Medicine, Yamagata University, Yamagata 990-9585, Japan; aaraki@med.id.yamagata-u.ac.jp (A.A.); h-nara@isenshu-u.ac.jp (H.N.); s-saitoh@med.id.yamagata-u.ac.jp (S.S.); asao-h@med.id.yamagata-u.ac.jp (H.A.); 2Department of Urology, Faculty of Medicine, Yamagata University, Yamagata 990-9585, Japan; ykurota11@yahoo.co.jp (Y.K.); kato-t@med.id.yamagata-u.ac.jp (T.K.); ito.hiromi@med.id.yamagata-u.ac.jp (H.I.); ntsuchiya@med.id.yamagata-u.ac.jp (N.T.); 3Department of Biological Sciences, Faculty of Science and Engineering, Ishinomaki Senshu University, Ishinomaki 986-8580, Japan; 4Division of Cancer Biology and Therapeutics, Miyagi Cancer Center Research Institute, Natori 981-1293, Japan; tanaka@med.tohoku.ac.jp

**Keywords:** IL-1β, NF-κB, oxidative stress, pantetheinase, prostate cancer cells

## Abstract

Recent studies have discovered a relationship between glycosylphosphatidylinositol (GPI)-anchored protein 80 (GPI-80)/VNN2 (80 kDa GPI-anchored protein) and malignant tumors. GPI-80 is known to regulate neutrophil adhesion; however, the action of GPI-80 on tumors is still obscure. In this study, although the expression of GPI-80 mRNA was detectable in several tumor cell lines, the levels of GPI-80 protein were significantly lower than that in neutrophils. To clarify the function of GPI-80 in tumor cells, GPI-80-expressing cells and GPI-80/VNN2 gene-deleted cells were established using PC3 prostate cancer cells. In GPI-80-expressing cells, GPI-80 was mainly detected in vesicles. Furthermore, soluble GPI-80 in the conditioned medium was associated with the exosome marker CD63 and was also detected in the plasma obtained from prostate cancer patients. Unexpectedly, cell adhesion and migration of GPI-80-expressing PC3 cells were not modulated by anti-GPI-80 antibody treatment. However, similar to the GPI-80 family molecule, VNN1, the pantetheinase activity and oxidative state were augmented in GPI-80-expressing cells. GPI-80-expressing cells facilitated non-adhesive proliferation, slow cell proliferation, NF-κB activation and IL-1β production. These phenomena are known to be induced by physiological elevation of the oxidative state. Thus, these observations indicated that GPI-80 affects various tumor responses related to oxidation.

## 1. Introduction

Molecular targeted drugs and immune checkpoint inhibitors have opened a new era of cancer therapy. Recently, clinical studies have focused on the individual differences in therapeutic efficacy as well as the side effects [1]. Although the molecular mechanisms of resistance to these therapies have been elucidated, effective strategies to prevent or treat resistant tumors are nevertheless required to improve clinical outcomes for patients [2]. Therefore, in order to develop effective treatments, investigation into the mechanism of malignant transformation is the need of the hour.

Various comprehensive analytical studies have identified many molecules associated with tumor formation and malignancy. Recently, glycosylphosphatidylinositol (GPI)-anchored protein (GPI-80 also known as VNN2) has been identified as one of the molecules associated with chemotherapeutic resistance of tumors [3]. GPI-80, identified in human neutrophils [4], has been reported to regulate neutrophil adhesion and migration [4]. Furthermore, GPI-80 co-localizes with β2 integrin on resting neutrophils and is concentrated on pseudopodia during neutrophil migration [5]. Based on these observations, GPI-80 is considered to be a molecule that regulates adhesion on neutrophils. However, GPI-80 was detected not only on pseudopods but also in secretory vesicles [6], and soluble GPI-80 is released from activated neutrophils [7]. Besides neutrophils, hematopoietic stem cells also express GPI-80. Therefore, GPI-80 is considered to be a marker for hematopoietic stem cells.

Moreover, it has also been reported that GPI-80 is associated with malignant tumors [8]. In a previous study, microRNA-106a targeted GPI-80 and suppressed its expression. Inhibition of microRNA-106a induced overexpression of GPI-80 leading to reduced proliferation of osteosarcoma cells and increased apoptosis. This result suggested that GPI-80 might reduce osteosarcoma tumorigenesis. However, it has been reported that GPI-80 mRNA expression was upregulated in androgen-deprived prostate cancer cells (PC3 cells) [9] as well as in the metastatic human esophageal squamous cell carcinoma cell line (T.Tn-AT1 cells) [10]. These observations suggest that increased levels of GPI-80 may be involved in tumor growth and metastasis.

GPI-80 is a member of the pantetheinase gene family that consists of pantetheinase/VNN1, which produces pantothenic acid (vitamin B5) and cysteamine from pantetheine [11]. VNN1 is thought to limit the Warburg effect and suppress the growth of sarcomas [12]. It is still uncertain whether GPI-80 exhibits pantetheinase activity.

The present study examines the role of GPI-80 in tumor cells by overexpressing GPI-80 as well as deleting GPI-80 in PC3 cells. This study demonstrates the basic mechanism of action of GPI-80 in tumors.

## 2. Results

### 2.1. GPI-80 Expression Is Detected in Several Urologic Cancer Cell Lines, but the Expression Level Is Lower Than That of Neutrophils

Previous reports have described an association between tumor malignancy and GPI-80 levels [3,8,9,10]. However, there is no mouse homolog of GPI-80/VNN2, and therefore, it is difficult to investigate its function using mouse models. To examine the function of GPI-80 in tumors, this study investigated its expression levels in representative urinary cancer cell lines, (three prostate cancer cell lines (PC3, DU145, and LNCaP), two kidney cancer cell lines (A-704 and Caki-1), and four bladder cancer cell lines (HT1376, RT-4, SCaBER, and T-24)). Among these cell lines, PC3 cells (prostate cancer cell line) and A-704 cells (kidney cancer cell line) expressed relatively higher levels of GPI-80 mRNA than other cells (Appendix A). In PMN leukocytes, the expression of both full-length GPI-80 mRNA and alternative-spliced form of GPI-80 mRNA was clearly observed using the same primer set. However, the expression of the alternative-spliced form of GPI-80 mRNA was not observed in these tumor cell lines (Appendix A). In the CHO transformant using the full-length cDNA of GPI-80, expression of alternative-spliced form of GPI-80 mRNA was not detected (Appendix A).

Neither mRNA nor flow cytometric analysis showed a correlation between cell line malignancy and spontaneous GPI-80 expression levels. In particular, RT-4 is famous as a low-grade urothelial carcinoma cell line, and T24 is famous as a high-grade urothelial carcinoma cell line. The spontaneous expression level of GPI-80 did not change between RT-4 and T-24 (Appendix A).

When the protein levels of GPI-80 in these cell lines were analyzed by flow cytometry, they were significantly lower than those in neutrophils (Appendix A). Among these tumor cell lines, PC3 cells had relatively higher GPI-80 levels than other cell lines (Appendix A). Therefore, in this study, PC3 cells were used to investigate the function of GPI-80 in tumor cells.

### 2.2. Establishment of GPI-80-Overexpressing PC3 Cells and GPI-80 Gene-Deleted PC3 Cells

Previously, neutrophils altered the cell adhesion ability upon treatment with anti-GPI-80 mAb (3H9) [4]. As described above, PC3 cells seemed to be better than other cell lines for investigating the function of GPI-80. However, the GPI-80 level in PC3 cells was not detected by western blotting (Appendix A). Therefore, to analyze the function, a GPI-80-expressing PC3 cell clone was established, which was named as #22. Subsequently, the GPI-80 gene and GPI-80 cDNA in #22 were deleted using the lentivirus-CRISPR/Cas9 system, and the cell clone was named as #22ΔGPI-80. The guide RNA sequence of CRISPR/Cas9 could bind to both the transfected GPI-80 cDNA and GPI-80 gene in the genome. For the control, #22 was infected with mock (without the guide RNA sequence) lentivirus-CRISPR/Cas9 and was named as #22mock. GPI-80 levels in these cells were confirmed by western blotting and flow cytometry (Appendix A).

Attempts to establish GPI-80-expressing cells using other cell lines were tried. T-24 cells and RT-4 cells also expressed GPI-80 mRNA (Appendix A). Therefore, these cell lines were infected with the lentiviral vector. However, these transient GPI-80-expressing cells accounted for less than 10% after GPI-80 transfection, and the number of GPI-80-expressing cells did not increase with drug selection (Appendix A). When PC3 cells were used for transient expression of GPI-80, approximately 40% of GPI-80-positive cell subset was obtained. Thus, it was decided to use #22mock and #22ΔGPI-80 cells for this study.

### 2.3. GPI-80 Localized in Vesicles and Was Detected in Conditioned Medium with Exosome Marker, CD63

The subcellular localization of GPI-80 was next investigated to understand the function of GPI-80. The GPI-80 expression was examined by confocal microscopy. In this study, GPI-80 cDNA was fused to a FLAG tag sequence, which was further fused to a signal peptide sequence. As a result, the GPI-80 level could be detected using both PE-conjugated anti-GPI-80 antibody (3H9) and FITC-conjugated anti-FLAG antibody. Overlap antibody reactions detected GPI-80 as yellow color. CD29 (also known as integrin β1), which is an adhesion marker for epithelial cells, was detected using APC-conjugated anti-CD29 antibody as pink color. CD29 level was abundantly detected on the cell surface, while the co-localization of CD29 with FLAG-GPI-80 was not clearly observed (Figure 1a). Conversely, FLAG-GPI-80 was mainly observed in the vesicle (Figure 1a). Because the cells were stained without cell membrane-penetrating treatment, it was assumed that FLAG-GPI-80 was localized in secreted extracellular vesicles (EVs) rather than intracellular vesicles. To confirm the localization of GPI-80 in EVs, the cells were stained with FITC-conjugated Annexin V to detect the externalized phosphatidylserine, which is one of the markers for EVs [13,14,15]. As a result, GPI-80 was co-localized with externalized phosphatidylserine detected using Annexin V (Figure 1b). As a negative control, # 22ΔGPI-80 cells were stained using the same method, but no vesicles were observed (Figure 1c).

Next, in order to confirm GPI-80 localization at the tip of the pseudopod and filopodium, F-actin was stained with phalloidin after cell membrane permeabilization using 0.5% Triton-X 100. In addition, to clarify the localization of GPI-80, the signal was enhanced using indirect antibody staining using rabbit F(ab’)2 immunoglobulin FITC-conjugated anti-mouse IgG antibody (Figure 1d). As a result, GPI-80 localized at the tip of the pseudopod and filopodium, and the accumulation of GPI-80 in the large vesicle was clearly observed (Figure 1d). Unexpectedly, GPI-80 localized at the tip of the pseudopod and filopodium was detected as a vesicle (Figure 1d), suggesting that GPI-80 was mainly accumulated in vesicles and secreted in EVs from PC3 cells.

To verify GPI-80 secretion in EVs, soluble GPI-80 (sGPI-80) was measured in the plasma of prostate cancer patients and in the conditioned medium of PC3 cells expressing GPI-80 using sandwich ELISA with two types of anti-GPI-80 mAb clone, 3H9 (IgG1, used for capture) and 4D4 (IgG2a, used for detection). As expected, the relative amounts of sGPI-80 in the plasma of prostate cancer patients were higher than that from healthy volunteers (Figure 1e). Furthermore, sGPI-80 was detected in the conditioned medium from #22mock cells but not #22ΔGPI-80 cells (Figure 1f). CD63 is a marker for EVs, and therefore, the authors examined the association of sGPI-80 with CD63 using sandwich ELISA, wherein sGPI-80 was trapped using anti-GPI-80 mAb (3H9) and detected using anti-CD63 mAb (IgG2a) (Figure 1g). The association of sGPI-80 with CD63 was detected in the medium from #22mock but not #22ΔGPI-80 cells (Figure 1g). Furthermore, it was detected that sGPI-80 associated with CD63 in the conditioned medium of GPI-80-expressing HEK293T cells (Appendix A). These results indicated that GPI-80 was located in the secretory vesicles of PC3 cells and released in EVs.

### 2.4. GPI-80 Tends to Undergo Intracellular Oxidization

Pantetheinase hydrolyzes pantetheine to produce pantothenic acid and cysteamine. Cysteamine is known to inhibit glutathione synthesis, resulting in intracellular oxidation. Therefore, the ratio of the amounts of GSH and GSSG was measured. There was no difference in the amount of GSH per total protein between #22mock and #22ΔGPI-80 cells (Figure 2a, left panel), but the amount of GSSG in #22mock cells was higher than that in #22ΔGPI-80 cells (Figure 2a, middle panel). Therefore, the ratio of the amounts of GSH and GSSG in #22mock cells was significantly lower than that in # 22ΔGPI-80 cells (Figure 2a, right panel). In contrast, there was no difference in the amount of GSH in the culture media from #22mock and #22ΔGPI-80 cells (Figure 2b). These results suggest that GPI-80 tends to undergo intracellular oxidization.

### 2.5. GPI-80 Expression Does Not Augment the Growth of PC3 Cells

GPI-80 expression was detected in self-renewing hematopoietic stem cells and malignant tumors [3,8,16]. These observations allowed the speculation that GPI-80 expression might augment cell growth or colony formation. First, cell growth on the culture plate was examined. The growth of GPI-80-expressing cells (#22mock) was lower than that of parental PC3 cells. In addition, the growth of GPI-80 gene-deleted cells (#22ΔGPI-80) was also lower than that of PC3 cells (Figure 3a). These results suggested that GPI-80 expression was not related to PC3 cell growth. Anchorage independent growth is a critical step in the tumorigenic transformation of cells [17]. Thus, cell growth on agarose was assayed to clarify the relationship between GPI-80 expression and cell survival under floating conditions. The growth of GPI-80-expressing cells (#22mock) was not augmented on agarose, as compared with that of PC3 cells (Figure 3b). However, the growth of GPI-80 gene-deleted cells (#22ΔGPI-80) was lower than that of both PC3 and #22mock cells (Figure 3b). These results suggested that GPI-80 expression might support cell survival under floating conditions. Next, colony formation ability was checked. The number of colonies formed by #22mock cells was slightly augmented, but not significantly increased, and the colony formation ability of #22ΔGPI-80 cells did not change (Figure 3c). These observations indicated that, while GPI-80 expression did not augment cell growth, it might have had a supportive function in cell survival under floating conditions.

It has been reported that redox potential regulates cell proliferation. Active cell proliferation requires a reducing state, of approximately −260 mV, and slow cell proliferation needs a slightly oxidized state, of approximately −220 mV [18,19]. When cells are treated with a low dose of reducing agent, such as *N*-acetylcysteine, the difference in reducing potential between the extracellular and intracellular environments is decreased owing to disulfide breaking activity [18,20]. However, treatment with a high dose of reducing reagent causes a reducing environment in the intracellular space. Indeed, addition of low dose *N*-acetylcysteine augments the sensitivity to reactive oxygen [21]. In this study, the proliferation of GPI-80-expressing cells was significantly suppressed by the addition of low-dose *N*-acetylcysteine (0.3125 mM) (Figure 3d). These results suggested that the proliferation of GPI-80-expressing cells was sensitive to extracellular redox balance.

### 2.6. GPI-80 Augments NF-κB Activation

NF-κB is an intracellular signal transducer whose activation is dependent on the oxidative state [22]. NF-κB activation undergoes cycles of cytoplasmic-to-nuclear and nuclear-to-cytoplasmic transport, giving rise to so called “oscillations,” which regulate the transcriptional output [23,24]. A small fraction of unstimulated cells showed spontaneous random activation of NF-κB, and a fraction of responding cells did not show any oscillation [24].

To estimate the effect of oxidation induced by GPI-80 expression, the authors investigated whether GPI-80 expression is involved in NF-κB activation. To detect the levels of activated NF-κB in a cell, phosphorylated p65-NF-κB level was measured by flow cytometry. As shown in (Figure 4a), the levels of phosphorylated p65-NF-κB in #22ΔGPI-80 cells were lower than that in #22mock cells. This result suggested that GPI-80 expression is involved in the augmentation of spontaneous NF-κB activation.

NF-κB is one of the most important regulators of proinflammatory gene expression and regulates a positive autoregulatory loop for IL-1β level [25,26]. To verify the involvement of GPI-80 expression in the functional activation of NF-κB, the response to IL-1β production using #22mock and #22ΔGPI-80 cells was assayed. Although there was no significant difference in the levels of IL-1β production between #22mock and #22ΔGPI-80 cells, the levels of IL-1β production seemed to be reduced in #22ΔGPI-80 cells. Indeed, a significant increase in IL-1β level was detected in #22mock cells upon stimulation with PMA and LPS, while there was no increase in IL-1β level in #22ΔGPI-80 cells (Figure 4b). This observation suggested that GPI-80 expression augmented the sensitivity to functional NF-κB activation.

To confirm the involvement of GPI-80 in NF-κB activation, GPI-80 was transiently expressed in PC3 cells. The percentage of phosphorylated p65-NF-κB was increased in the GPI-80^+^ cell subset of GPI-80-transfected cells as compared with that in mock-transfected cells (Figure 5). The level of NF-κB activation in GPI-80^+^ cell subset was significantly higher than that in both GPI-80^-^ cell subset and mock-transfected cells. The reproducibility of the NF-κB activation in the GPI-80^+^ cell subset was confirmed using other cell lines, HEK293T and T24 cells (Appendix A). These results indicated that GPI-80 expression augmented NF-κB activation in PC3 cells.

### 2.7. GPI-80 Expression Does Not Control Cell Adhesion and Migration in PC3 Cells

The relationship between GPI-80 expression and tumor migration has been previously reported [8,10]. To verify the effect of GPI-80 on PC3 cell adhesion and migration, GPI-80-expressing cells (before cloning of GPI-80-expressing PC3 cells, PC3-GPI-80 oligo-clone, and #22mock), mock-infected PC3 cells (PC3-mock oligo clone), and GPI-80 gene-deleted cells (#22ΔGPI-80) were used for these assays. The ability of these cells to adhere and migrate was not modulated upon anti-GPI-80 mAb treatment, even in GPI-80-expressing cells (Appendix A). These results indicated that GPI-80 expression did not have any regulatory function on cell adhesion and migration in PC3 cells.

### 2.8. GPI-80 Has Weak Pantetheinase Activity in PC3 Cells

GPI-80 belongs to the VNN1/pantetheinase family. To verify the enzymatic activity of GPI-80 in PC3 cells, pantetheinase activity using pantothenate-AMC was measured.

First, the activities of VNN1 and GPI-80 were compared. To show the similar fluorescence value, recombinant VNN1 (53 kDa; 0.125 μg/sample was approximately 2.36 pmol) and sGPI-80/Fc (approximately 100 kDa under reducing conditions; 1 μg/sample was approximately 10 pmol) were used. The molecular mass of sGPI-80/Fc, which was GPI-80 fused with immunoglobulin Fc fragment, was confirmed by immunoblot analysis (Appendix A). The pantetheinase activity was detected in the purified sGPI-80/Fc, and the estimated activity of GPI-80 per molecule was 4.2 times lower than that of VNN1 (Appendix A). The pantetheinase activity was measured in the cell lysates of GPI-80-expressing cells (#22mock) and GPI-80 gene-deleted cells (#22ΔGPI-80). The pantetheinase activity in the cell lysate of #22mock cells was slightly higher than that in the cell lysate of #22ΔGPI-80 cells, although the pantetheinase activity was not abolished in #22ΔGPI-80 (Appendix A). Surprisingly, a remarkably high pantetheinase activity was observed in FCS-containing cell culture medium (Appendix A). Therefore, it was difficult to estimate the pantetheinase activity of GPI-80 derived from the conditioned medium. Furthermore, the pantetheinase activity in the plasma from healthy volunteers was higher than that of sGPI-80/Fc (Appendix A). These observations suggested that the pantetheinase activity of GPI-80 was weak compared to the activity in extracellular fluid.

## 3. Discussion

GPI-80 promoted non-adhesive proliferation, slow cell proliferation, and IL-1β production in PC-3 cells. Especially, NF-κB activation was facilitated in the GPI-80^+^ cell subset. Furthermore, the secreted soluble GPI-80 from PC3 cells was co-localized with exosome markers, and soluble GPI-80 was detected in the plasma of high-risk group prostate cancer patients. These observations suggested that GPI-80 might diffuse and thereby play a role in the formation of tumor microenvironment.

In recent years, expression of GPI-80 has been found that its expression level may be negatively correlated in survival of cancer patients (The human protein atlas: https://www.proteinatlas.org/ENSG00000112303-VNN2/pathology/renal+cancer; last accessed the link, 6 November 2021). In this study, the function of GPI-80 in tumor cells is thought to induce the release of sGPI-80 and the activation of NF-κB. Because cancer-induced chronic inflammation is known to suppress the immune response [27], GPI-80 expression may induce chronic inflammation and reduce survival. In the future, sGPI-80 released into the blood may be a useful index for examining chronic inflammation and immunosuppression in cancer patients.

Among the multiple tumor cells that were examined, PC-3 cells were able to confirm the most stable expression level of GPI-80 by flow cytometric analysis. On the other hand, PC-3 cells have been used in adhesion tests and migration experiments [28,29]. From these facts, it was assumed that PC-3 cells are suitable for studying the functions of GPI-80 for adhesion and migration. Disappointedly, no clear adhesion control ability of GPI-80 was observed even with PC-3 cells (Appendix A, PC-3 mock oligo clone). Overexpressing of a target molecule in any cells is commonly used to clarify the molecular function. Unfortunately, among the human cell lines that were examined, only PC-3, HEK293T, and T-24 could stably overexpress GPI-80 (Appendix A). Therefore, in this study, GPI-80 overexpressing PC-3 cells and the cells in which GPI-80 expression was deleted from GPI-80 overexpressing PC-3 cells were used to clarify the function of GPI-80. 

GPI-80, which is often expressed in malignant tumors, is known to regulate neutrophil adhesion and migration [4]. However, in this study, neither mRNA nor flow cytometric analysis showed a correlation between cell line malignancy and spontaneous GPI-80 expression levels (Appendix A). Furthermore, GPI-80 expression did not affect the adhesion and migration of PC3 cells (Appendix A). Thus, it was concluded that the adhesion controlling ability of GPI-80 is cell type-specific.

Oxidative conditions are appropriate for tumorigenesis. For example, myeloid cell-derived reactive oxygen species induce tumor progression and initiation [30], and GSSG-accumulated macrophages reduce IL-12 production [31]. It is also known that the GSSG/GSH ratio regulates both the second phase of neutrophil-endothelial cell adhesion and prostate cancer cell invasion [32,33]. The oxidation induced by pantetheinase activity is known to inhibit γ-glutamylcysteine synthase activity via cysteamine synthesis [11,12]. This current study detected weak pantetheinase activity (Appendix A) and an increase in the levels of GSSG in GPI-80-expressing cells (Figure 2). One possibility is that GPI-80 levels in tumor cells might be associated with oxidative conditions in the tumor microenvironment.

The direct effects of the oxidative state in PC3 cells indicate that the oxidized state of Src prevents apoptosis in the absence of adhesion [34]. In this study, the proliferation of PC3 cells without adhesion (on agarose) was reduced by deletion of the GPI-80 gene. These results suggested that GPI-80 may favor survival of cells under floating conditions, such as of circulating tumor cells.

Additionally, proliferating cells are maintained in a reduced state. When cell proliferation is active, the redox potential is maintained in a reduced state (−260 mV); when the cells exhibit slow proliferation, they are in a slightly oxidized state (−220 mV) [18]. The maintenance of mesenchymal stem cells or pluripotent stem cells requires physiological levels of reactive oxygen species as second messengers [19]. Interestingly, GPI-80 level was selectively detected in self-renewing hematopoietic stem cells [16]. Similar to stem cells, a distinct subpopulation of slow-cycling melanoma cells is required for continuous tumor growth [35]. Indeed, GPI-80-induced growth of GPI-80-expressing PC3 cells was slightly but significantly reduced. These observations suggested that GPI-80 might be involved in the growth of slow-proliferating subpopulation of cells in tumors.

An inflammatory tumor microenvironment plays a key role in driving tumor initiation, growth, progression, and metastasis [27]. NF-κB activation, which induces inflammation, is augmented in GPI-80-expressing cells. It is also known that NF-κB activation is facilitated by oxidative conditions [22]. These observations suggested that GPI-80 expression or secreted GPI-80 contributes to an inflammatory tumor microenvironment.

Extracellular redox modulation can regulate immune responses and tumor cell proliferation [18,36]. As GPI-80 is a GPI-anchored protein, via the enzymatic activity of GPI-80, cysteamine is synthesized on the cell membrane as an ectoenzyme. As a result, cysteamine can break disulfide bonds [11] and modify the extracellular redox balance similar to thioredoxin. In conclusion, GPI-80 level in tumor cells is associated with several events such as survival in non-adherent conditions, slow cell proliferation, or protumorigenic inflammation.

## 4. Materials and Methods

### 4.1. Human Peripheral Blood and Plasma

All methods were performed in accordance with the relevant guidelines and regulations. All healthy volunteers signed an informed consent prior to blood collection. This study was approved by the Ethics Committee of the Yamagata University Faculty of Medicine (approval numbers: H28-265,5 October 2016; H29-101, 12 June 2017). The subjects of this study included four patients with biopsy-diagnosed prostate cancer at the Department of Urology in the Yamagata University Hospital between April 2017 and January 2019. The mean age of the patients was 69.5 ± 5.0 years. All patients had prostate cancer with a Gleason score of 8 or higher and were classified in the high-risk group according to the National Comprehensive Cancer Network (NCCN) risk classification. Appropriate treatment was provided to each patient according to their medical condition.

The average age of healthy volunteers was 47.0 ± 9.2 years (n = 4, male). Polymorphonuclear leukocytes (PMNs) were purified from peripheral blood samples of volunteers using Dextran and Ficoll-Paque as described previously [37].

### 4.2. Measurement of Reduced Glutathione (GSH) and Oxidized Glutathione (GSSG)

Confluent cells (incubated for 3 days) were washed with PBS and collected using a cell scraper. The conditioned medium was collected at the same time. The cells (0.5 − 1 × 10^7^ cells) were suspended in 10 mM HCl (80 μL/sample), and then freeze-thawed (repeated two times). After centrifugation (1000× *g*, 10 min at 4 °C), the supernatants or the conditioned medium were mixed with 5% 5-sulfosarityl acid (20 μL/sample or equal volume/medium, respectively). The samples were stored at −80 °C until the assay was performed. GSH and GSSG levels were quantified according to the manufacturer’s protocol (GSSG/GSH Quantification Kit; Dojindo, Kumamoto, Japan). The protein concentration of each sample was determined using the BCA protein assay kit (Takara Bio, Kusatsu, Japan). The absorbance was measured at 405 nm using a microplate reader (Sunrise Remote, TECAN, Männedorf, Switzerland).

### 4.3. Measurement of Phosphorylated p65-NF-κB

Staining of cells to detect phosphorylated p65-NF-κB was performed as described previously [37]. In brief, the cells were fixed with Phosflow Lyse/Fix buffer (BD Biosciences, San Jose, CA, USA) for 10 min at 37 °C. After fixation, the cells were suspended in ice-cold 90% methanol for membrane permeabilization. After permeabilization, the cells were washed with PBS containing 3% FCS and 0.1% NaN3, and then incubated with Alexa 647-anti-pS536-NF-κB-p65 rabbit mAb (93H1; Cell Signaling Technology, Danvers, MA, USA) and PE-conjugated anti-GPI-80 mAb (3H9) for 30 min at 23–25 °C. After washing with PBS, the samples were stored at 4 °C.

### 4.4. Cell Growth Assay on Agarose and Colony Formation Assay in Soft Agarose

The bottom layer of 0.5% (*w*/*v*) agarose (0.5 mL/well) was prepared by mixing 5% agarose (50 °C) with culture medium (37 °C) and incubating at 4 °C for 1 h. The cells (3 × 10^3^ cells/0.5 mL/well) were cultured in a 24-well culture plate or on 0.5% agarose for 10 days. After incubation, the number of cells was quantified using AlamarBlue™ (Thermo Fisher Scientific, Waltham, MA, USA). For the colony formation assay, the method described previously [38] was modified**.** The cells (3 × 10^3^) were suspended in 0.25% agarose (0.5 mL/well) and seeded on the bottom layer of 0.5% agarose (0.5 mL/well). The culture medium (0.5 mL/well) was added on top of the agar layer. The cells were cultured for 20 days, and the number of colonies (diameter > 0.4 mm) per well was counted using ImageJ version 1.43u.

### 4.5. IL-1β Measurement

The cells (1 × 10^5^ cells/mL) were incubated in the absence or presence of lipopolysaccharide (LPS, 10 μg/mL; Sigma-Aldrich, St. Louis, MO, USA) plus phorbol 12-myristate 13-acetate (PMA, 1 ng/mL; Sigma-Aldrich) for 24 h. After incubation, the conditioned medium was collected and centrifuged at 10,000× *g* for 1 min at 4 °C. The supernatants were stored at −80 °C until the assay. IL-1β levels in the samples were measured using the cytometric bead array-enhanced flex bead set by flow cytometry (FACSCanto II, BD Biosciences). The data were analyzed using FCAP Array software (version 1.0.1; BD Biosciences).

### 4.6. Statistical Analysis

All the data are displayed as scattered dots and mean values. The data indicating dose effects are presented as the mean ± standard error. Statistical analyses methods are indicated in each figure legend, and the calculations were performed using the GraphPad Prism software, version 5.03 (San Diego, CA, USA). Results with *p* values less than 0.05 were considered as statistically significant.

## Figures and Tables

**Figure 1 ijms-22-12027-f001:**
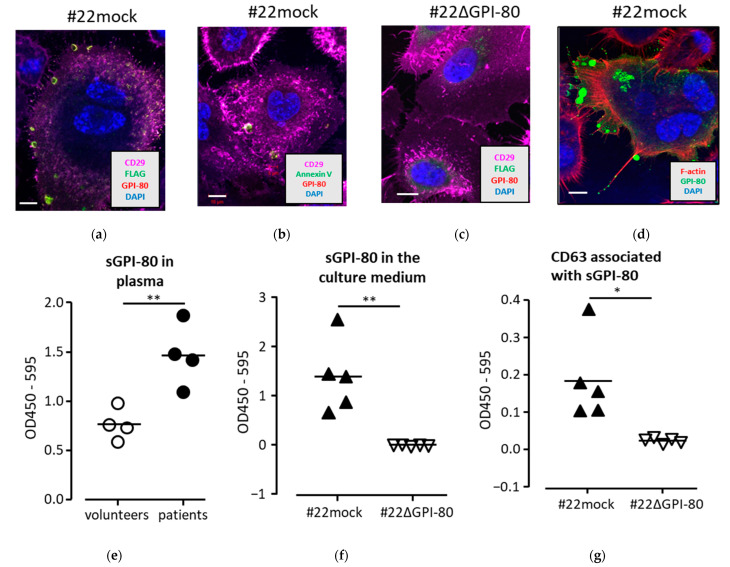
Localization of GPI-80 on PC3 transformant. (**a**–**d**) GPI-80 was observed by confocal microscopy. Cells (#22mock (**a**) and #22ΔGPI-80 (**c**)) were stained with anti-CD29 mAb (pink), anti-FLAG mAb (green), anti-GPI-80 mAb (yellow), and DAPI (blue). Cells (#22mock (**b**)) were also stained with anti-CD29 mAb (pink), anti-GPI-80 mAb (yellow), DAPI (blue), and Annexin V (green). (**d**) After fixation and incubation with unlabeled anti-GPI-80 mAb (3H9) and FITC-conjugated anti-mouse antibody (green), the permeabilized #22mock cells were stained with phalloidin (red) to detect F-actin. Data are representative of results from more than three independent experiments. Scale bar, 10 μm. (**e**–**g**) Detection of soluble GPI-80 by sandwich ELISA. (**e**) Soluble GPI-80 level in the plasma of healthy volunteers (four volunteers; open circle) and the prostate cancer patients (four patients; closed circle) was detected by paired anti-GPI-80 mAbs, 3H9 and 4D4. (**f**) Soluble GPI-80 levels in the conditioned medium of #22mock (closed triangle) and #22ΔGPI-80 (open triangle) cells were also measured using paired anti-GPI-80 mAbs. (**g**) To detect the colocalization of GPI-80 with CD63, conditioned media from #22mock (closed triangle) and #22ΔGPI-80 (open triangle) were analyzed using anti-GPI-80 mAb (3H9) paired with anti-CD63 mAb (8A12). These culture media were independently collected from confluent cells after 4 days. Statistical significance was calculated using two-tailed unpaired Student’s *t*-test (*, *p* < 0.05; **, *p* < 0.01).

**Figure 2 ijms-22-12027-f002:**
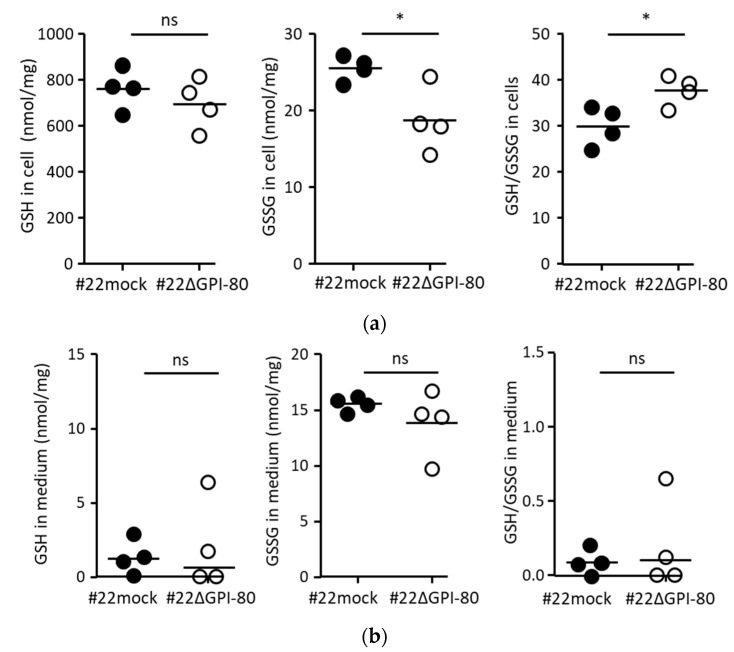
The ratio of GSH/GSSG was augmented in GPI-80-expressing cells. (**a**) The measurement of intracellular GSH/GSSG. The cells were cultured for 3 days, and approximately, 1 × 10^7^ cells were collected using a cell scraper. The intracellular GSH (left panel) and GSSG (middle panel) levels were measured using the cell lysate from #22mock (closed circle) and #22ΔGPI-80 (open circle) cells. The GSH and GSSG values (nmol/mg) were used for calculating the ratio of levels of GSH/GSSG (right panel). (**b**) The measurement of levels of GSH/GSSG in the conditioned medium. The cells were cultured for 3 days, and the media were collected. The levels of GSH (left panel) and GSSG (middle panel) were measured using the conditioned medium from #22mock (closed circle) and #22ΔGPI-80 (open circle) cells. The GSH and GSSG values (nmol/mg) were used for calculating the ratio of levels of GSH/GSSG (right panel). The data are representative of four independent experiments, and the statistical significance was calculated using two-tailed unpaired Student’s *t*-test (*, *p* < 0.05; ns, not significant).

**Figure 3 ijms-22-12027-f003:**
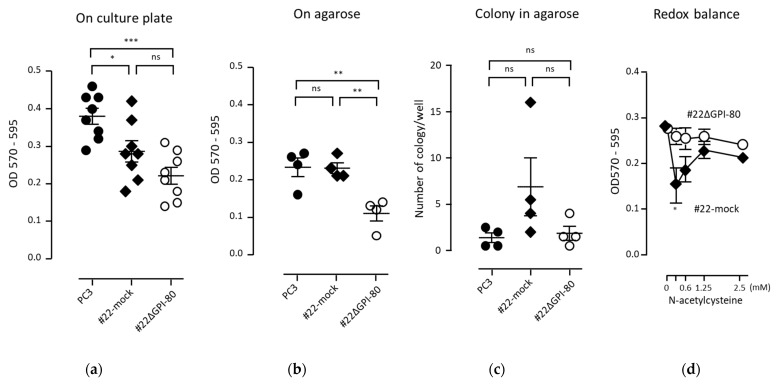
The effect of GPI-80 level on cell growth. PC3 cells (closed circle), #22 (closed diamond) and #22ΔGPI-80 (open circle) were cultured in various conditions. The cells (3 × 10^3^ cells/0.5 mL/well) were cultured in a 24-well culture plate (**a**), or on 0.5% agarose (**b**), for 10 days. For colony formation assay (**c**), the cells were cultured for 20 days. The data are representative of more than four independent experiments. The statistical significance (**a**–**c**) was calculated using one-way ANOVA with Bonferroni’s post hoc test, compared with each other (*, *p* < 0.05; **, *p* < 0.01; ***, *p* < 0.001; ns, not significant). To check the modulation of redox potential on cell growth (**d**), the cells (2.5 × 10^4^ cells/mL) were cultured in a 96-well plate for 3 days in the presence of *N*-acetylcysteine. The data are represented as the mean ± standard error from seven independent experiments. The statistical significance (**d**) was calculated using two-tailed unpaired Student’s *t*-test, compared between #22mock and #22ΔGPI-80 cells treated with similar dose of *N*-acetylcysteine (*, *p* < 0.05).

**Figure 4 ijms-22-12027-f004:**
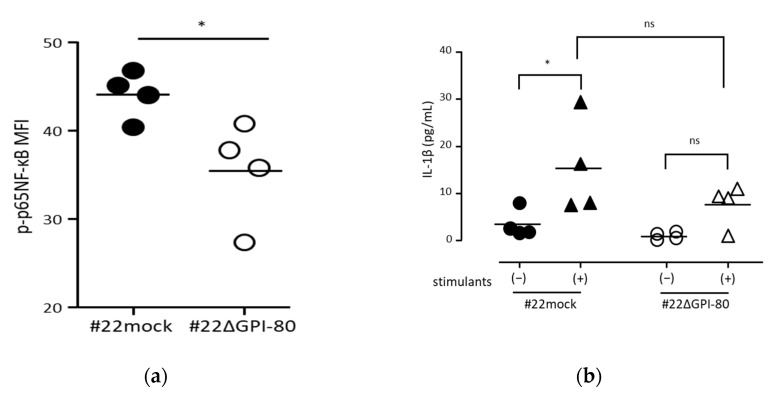
GPI-80 expression augmented NF-κB activation. (**a**) Measurement of phosphorylated p65-NF-κB levels. The levels of phosphorylated p65-NF-κB in #22mock (closed circle) and #22ΔGPI-80 (open circle) cells were measured by flow cytometry, and the mean fluorescence intensity (MFI) of phosphorylated p65-NF-κB (p-p65 NF-κB) was analyzed. The data represent results from four independent experiments. The statistical significance was calculated using two-tailed unpaired Student’s *t*-test (*, *p* < 0.05). (**b**) Measurement of IL-1β levels. Cells [#22mock (closed symbol) and #22ΔGPI-80 (open symbol)] were incubated with (stimulants (+), triangle symbol) or without (stimulants (−), circle symbol) 10 μg/mL LPS + 1 ng/mL PMA for 24 h. After incubation, the media were collected, and IL-1β concentration in the media was measured. The data represent results from four independent experiments, and the statistical significance was analyzed by one-way ANOVA with Bonferroni’s post hoc test, compared between: #22mock stimulants (−) and #22mock stimulants (+); #22ΔGPI-80 stimulants (−) and #22ΔGPI-80 stimulants (+); and #22mock stimulants (+) and #22ΔGPI-80 stimulants (+) (*, *p* < 0.05; ns, not significant).

**Figure 5 ijms-22-12027-f005:**
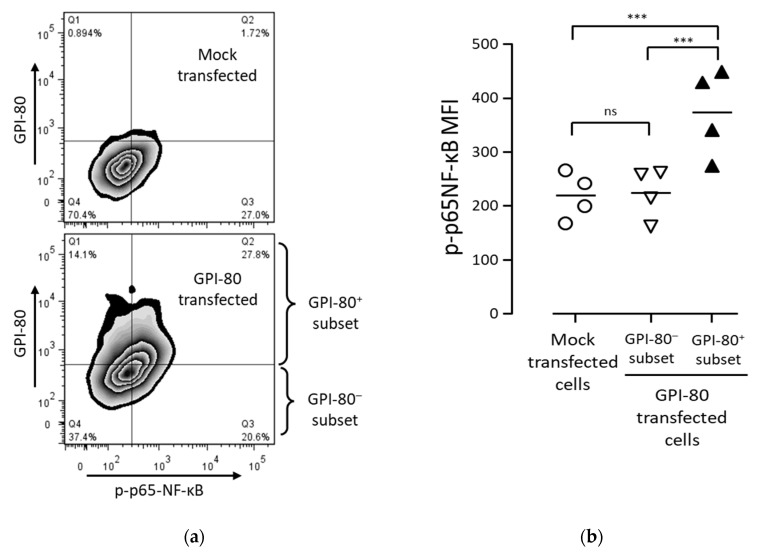
The relation between GPI-80-expressing cell subset and the activated NF-κB cell subset. (**a**) The representative flow cytometric analysis of GPI-80-expressing cell subset and activated NF-κB cell subset. PC3 cells were transfected with lentiviral vector packaged with pLV-SIN/puro-mock vector (upper panel) or pLV-SIN/puro-GPI-80 vector (lower panel), and the infected cells were incubated for 2 days. After incubation, the cells were stained with PE-conjugated anti-GPI-80 mAb (3H9) and Alexa 647-conjugated anti-phosphorylated p65-NF-κB mAb (93H1) and analyzed by flow cytometry. The representative analysis is from four independent experiments. The horizontal axis is phosphorylated p65-NF-κB (p-p65 NF-κB) level, and the vertical axis is GPI-80 level on the zebra-contour panels. (**b**) The p-p65 NF-κB MFI in mock-transfected PC3 cells. GPI-80^-^ cell subset and GPI-80^+^ cell subset were separated from GPI-80-transfected cells, as shown in (**a**). The p-p65 NF-κB MFI was analyzed in mock-transfected cells (open circle), GPI-80- cell subset (open inversed triangle) and GPI-80+ cell subset (closed triangle). The data represent results from four independent experiments, and the statistical significance was calculated by one-way ANOVA with Bonferroni’s post-hoc test, compared with each other (***, *p* < 0.001; ns, not significant).

## Data Availability

The data that support the findings of this study are available from the corresponding author, Y.T., upon reasonable request.

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
