# Peer review of "GPI-80 Augments NF-κB Activation in Tumor Cells"

_ijms, 2021, doi:10.3390/ijms222112027_

Round 1
Reviewer 1 Report
Takeda et al examined the role of GPI-80, a glycosylphosphatidylinositol -anchored protein, by overexpressing it in PC3 prostate cancer cells. They concluded that GPI-80 level was associated with the survival in floating conditions, proliferation and inflammation.
The experiments are well conducted, convincing and appropriate.
Nevertheless, GPI-80 is expressed in several urologic cancer cell lines as mRNA, but the protein level is low and not detectable in WB. For this reason, the authors decided to overexpress it in PC3 cells. My concern is that the results could be in some way speculative of a system that can not reproduce the reality.
Is there any correlation between the malignancy of the cell lines used in the study and the level of GPI-80?
I am convinced by the results as they are presented, but I would encourage the authors to better discuss the perspectives, the further studies and the outcomes of their discoveries in practical terms. They should highlight also the limit of the study. What it is missing is why the study is really scientific relevant to cancer therapy.
I would also use the supplemental Figure S5 as a normal Figure for the article. I think that the localization of GPI-80 as well as its level in human plasma and culture medium are important information.
I think that 7 self-citations are too much over 44 references. I would encourage the authors to reduce them.
Author Response
Response to reviewer-1
We thank you for your helpful comments. We found most of the comments valid and have helped us to strength our manuscript. We are grateful to you for your time. We have revised the manuscript precisely in line with the comments to the best of our ability. Our revisions are fully incorporated into the manuscript throughout and are also briefly itemized below.
(1) Reasons for using PC-3 cells and problem of reproduce
Among the multiple tumor cells we examined, PC-3 cells were able to confirm the most stable expression level of GPI-80 by flow cytometric analysis. On the other hand, PC-3 cells have been used in adhesion tests and migration experiments [28, 29]. From these facts, we assumed that PC-3 cells are suitable for studying the functions of GPI-80 for adhesion and migration. Disappointedly, no clear adhesion control ability of GPI-80 was observed even with PC-3 cells (Supplemental Figure S6a, PC-3 mock oligo clone). Overexpressing of target molecule in any cells is a commonly used to clarify the molecular function. Unfortunately, among the human cell lines we examined, only PC-3, HEK293T, and T-24 could stably overexpress GPI-80 (Supplemental Figure S3). Therefore, in this study, GPI-80 overexpressing PC-3 cells and the cells in which GPI-80 expression was deleted from GPI-80 overexpressing PC-3 cells were used to clarify the function of GPI-80. The re-producibility of the NF-κB activation in the GPI-80+ cell subset was confirmed using other cell lines, HEK293T and T24 cells (Supplemental Figure S5).
I have added these sentences to the discussion section (page 10, line 526-538).
(2) Is there a correlation between the malignancy of the cell line and the level of GPI-80?
Neither mRNA nor flow cytometric analysis showed a correlation between cell line malignancy and spontaneous GPI-80 expression levels. In particular, RT-4 is famous as a low-grade urothelial carcinoma cell line, and T24 is famous as a high-grade urothelial carcinoma cell line. The spontaneous expression level of GPI-80 did not change between on RT-4 and T-24 (Supplemental Figure S1).
The above points have been added to the result section 2.1 (page 2, lines 91-95).
And we have rewritten the discussion section as follow (page 10, lines 539-544):
GPI-80, which is often expressed in malignant tumors, is known to regulate neutro-phil adhesion and migration [4]. However, in this study, neither mRNA nor flow cytomet-ric analysis showed a correlation between cell line malignancy and spontaneous GPI-80 expression levels (Supplemental Figure S1). Furthermore, GPI-80 expression did not affect the adhesion and migration of PC3 cells (supplemental Figure S6). Thus, we concluded that the adhesion controlling ability of GPI-80 is cell type-specific.
(3) Discussion of prospects, further research, and outcomes
According to reviewer suggestion, we have rewritten the discussion as below (page 10, lines 512-525):
GPI-80 promoted non-adhesive proliferation, slow cell proliferation, and IL-1β pro-duction in PC-3 cells. Especially, NF-κB activation was facilitated in GPI-80+ cell subset. Furthermore, the secreted soluble GPI-80 from PC3 cells was co-localized with exosome markers, and soluble GPI-80 was detected in the plasma of high-risk group prostate cancer patients. These observations suggested that GPI-80 might diffuse and thereby play a role in the formation of tumor microenvironment.
In recent years, expression of GPI-80 has been found that its expression level may be negatively correlated in survival of cancer patients (The human protein atlas: https://www.proteinatlas.org/ENSG00000112303-VNN2/pathology/renal+cancer). In this study, the function of GPI-80 in tumor cells is thought to induce the release of sGPI-80 and the activation of NF-κB. Because cancer-induced chronic inflammation is known to suppress the immune response [27], GPI-80 expression may induce chronic inflammation and reduce survival. In the future, sGPI-80 released into the blood may be a useful index for examining chronic inflammation and immunosuppression in cancer patients.
(4) The localization of GPI-80 and its level in human plasma
Following the reviewer’s suggestion, the structure of the paper has been changed to Figure 1 instead of previous supplementary data.
Furthermore, adhesion test and pantheteisase activity measurement are moved to the second half of the results. (Suggested by another reviewer)
(5) reduce the references
Following the reviewer’s suggestion, we reduced the self-cited references and the similar refences.
Reviewer 2 Report
Although the article is really full of experiments, there are many things that need to be clarified and corrected, in my opinion. First of all it is not clear to me the reason of the chosen study model, since the GPI-80 protein is present at very low levels in PC-3 cells compared to neutrophils and in any case it does not control neither the adhesion nor the migration of the same cells.
The materials and methods aren't complete, the description of many techniques used is missing and I don't understand why most of the results inherent in the first part of the work are shown in the supplementary files. For these reasons, I suggest to reconsider after major changes.
Author Response
Response to reviewer-2
We thank you for your helpful comments. We found most of the comments valid and have helped us to strength our manuscript. We are grateful to you for your time. We have revised the manuscript precisely in line with the comments to the best of our ability. Our revisions are fully incorporated into the manuscript throughout and are also briefly itemized below.
(1) Reasons for overexpressing GPI-80 using PC-3 cells
Among the multiple tumor cells we examined, PC-3 cells were able to confirm the most stable expression level of GPI-80 by flow cytometric analysis. On the other hand, PC-3 cells have been used in adhesion tests and migration experiments [28, 29]. From these facts, we assumed that PC-3 cells are suitable for studying the functions of GPI-80 for adhesion and migration. Disappointedly, no clear adhesion control ability of GPI-80 was observed even with PC-3 cells (Supplemental Figure S6a, PC-3 mock oligo clone). Overexpressing of target molecule in any cells is a commonly used to clarify the molecular function. Unfortunately, among the human cell lines we examined, only PC-3, HEK293T, and T-24 could stably overexpress GPI-80 (Supplemental Figure S3). Therefore, in this study, GPI-80 overexpressing PC-3 cells and the cells in which GPI-80 expression was deleted from GPI-80 overexpressing PC-3 cells were used to clarify the function of GPI-80. The re-producibility of the NF-κB activation in the GPI-80+ cell subset was confirmed using other cell lines, HEK293T and T24 cells (Supplemental Figure S5).
I have added these sentences to the discussion section (page 10, line 526-538).
(2) Missing of technical explanation
In this experiment, cell line establishment and Fc fusion protein production will take a very large space, so the main points are narrowed down and described in Supplemental Materials and Methods. Adhesion test [supplemental references 6] and pantetheine assay [supplemental references 8] were previously described in “Legend”. In revised paper, the references are listed, and these methods are described in the supplemental Materials and Methods.
Please check the revised “Supplemental Materials and Methods”.
(3) The reason why most of the results are shown in the supplementary file in the first part
In this study, there are many newly created experimental materials. As you pointed out, this is an irregular paper. Due to space limitations for publishing papers, it was necessary to shorten it, so it became many supplementary data. The supplemental results in this study do not indicate novel function of GPI-80. However, these supplementary data are important in considering the function of GPI-80 in tumor cells. Thus, according to your suggestions, we have revised the structure of the paper as follows:
1) Figure 1 (previous supplemental Figure S5) shows the localization of GPI-80 and the results of being released in the extracellular vesicle (pointed out by another reviewer).
2) Adhesion test and pantheteisase activity measurement are moved to the second half of the results.
3) Furthermore, we have rewritten the discussion to find the citation of supplemental data as belows:
- i) page 10, line 539-544
GPI-80, which is often expressed in malignant tumors, is known to regulate neutrophil adhesion and migration [4]. However, in this study, neither mRNA nor flow cytometric analysis showed a correlation between cell line malignancy and spontaneous GPI-80 expression levels (Supplemental Figure S1). Furthermore, GPI-80 expression did not affect the adhesion and migration of PC3 cells (supplemental Figure S6). Thus, we concluded that the adhesion controlling ability of GPI-80 is cell type-specific.
- ii) page 11, line 568-573
The oxidation induced by pantetheinase activity is known to inhibit γ-glutamylcysteine synthase activity via cysteamine synthesis [11, 12]. In this study, we detected weak pantetheinase activity (Supplemental Figure S8) and an increase in the levels of GSSG in GPI-80-expressing cells (Figure 2). One possibility is that GPI-80 levels in tumor cells might be associated with oxidative conditions in the tumor microenvironment.
Please check the revised manuscript.
Round 2
Reviewer 2 Report
The authors have improve the manuscript and answer to all my observations. For these reasons, it can be accepted in present form.